# Multi-strain probiotics (Hexbio) containing MCP BCMC strains improved constipation and gut motility in Parkinson's disease: A randomised controlled trial

Azliza Ibrahim[1]☯, Raja Affendi Raja Ali[2], Mohd Rizal Abdul Manaf[3], Norfazilah Ahmad[3], Farah Waheeda Tajurruddin[4], Wong Zhi Qin[2], Siti Hajar Md Desa[5], Norlinah Mohamed Ibrahim[1]☯*

1 Faculty of Medicine, Neurology Unit, Department of Medicine, Hospital Canselor Tuanku Muhriz, UKM Medical Centre, Kuala Lumpur, Malaysia, 2 Faculty of Medicine, Gastroenterology Unit, Department of Medicine, Hospital Canselor Tuanku Muhriz, UKM Medical Centre, Kuala Lumpur, Malaysia, 3 Faculty of Medicine, Department of Community Health, Hospital Canselor Tuanku Muhriz UKM Medical Centre, Kuala Lumpur, Malaysia, 4 Department of Pharmacy, Hospital Canselor Tuanku Muhriz, UKM Medical Centre, Kuala Lumpur, Malaysia, 5 Department of Nursing, Hospital Canselor Tuanku Muhriz, UKM Medical Centre, Kuala Lumpur, Malaysia

☯ These authors contributed equally to this work.
* norlinah@ppukm.ukm.edu.my, norlinah@gmail.com

## Abstract

### Objective

We determined the effectiveness of a multi-strain probiotic (Hexbio®) containing microbial cell preparation MCP®BCMC® on constipation symptoms and gut motility in PD patients with constipation.

### Methods

PD patients with constipation (ROME III criteria) were randomized to receive a multi-strain probiotic (*Lactobacillus sp* and *Bifidobacterium sp* at 30 X 10⁹ CFU) with fructo-oligosaccaride or placebo (fermented milk) twice daily for 8 weeks. Primary outcomes were changes in the presence of constipation symptoms using 9 items of Garrigues Questionnaire (GQ), which included an item on bowel opening frequency. Secondary outcomes were gut transit time (GTT), quality of life (PDQ39-SI), motor (MDS-UPDRS) and non-motor symptoms (NMSS).

### Results

Of 55 recruited, 48 patients completed the study: 22 received probiotic and 26 received placebo. At 8 weeks, there was a significantly higher mean weekly BOF in the probiotic group compared to placebo [SD 4.18 (1.44) vs SD 2.81(1.06); (mean difference 1.37, 95% CI 0.68, 2.07, uncorrected *p*<0.001)]. Patients in the probiotic group reported five times higher odds (odds ratio = 5.48, 95% CI 1.57, 19.12, uncorrected *p* = 0.008) for having higher BOF (< 3 to 3–5 to >5 times/week) compared to the placebo group. The GTT in the probiotic

**Data Availability Statement:** All data are uploaded into a public repository and fully accessible at: https://doi.org/10.5061/dryad.np5hqbzr7.

**Funding:** NMI received funding from Universiti Kebangsaan Malaysia Research Grant (FF-2018-387) The active treatment and placebo were provided free of charge by study sponsor (B-CROBES SDN BHD). The funders had no role in study design, data collection and analysis, decision to publish or in the preparation of the manuscript.

**Competing interests:** The probiotics and the placebo were provided by BCROBES SDN BHD. B-CROBES had no role in the study design, data collection and data analysis. On behalf of the authors, I declare that there are no competing interests, personal, professional, financial or nonfinancial related to the study or its publication. This does not alter our adherence to PLOS ONE policies on sharing data and materials. There are no restrictions on sharing of data and/or materials related to this study.

group [77.32 (SD55.35) hours] reduced significantly compared to placebo [113.54 (SD 61.54) hours]; mean difference -36.22, 95% CI -68.90, -3.54, uncorrected $p = 0.030$). The mean change in GTT was 58.04 (SD59.04) hour vs 20.73 (SD60.48) hours respectively (mean difference 37.32, 95% CI 4.00, 70.63, uncorrected $p = 0.028$). No between-groups differences were observed in the NMSS, PDQ39-SI, MDS-UPDRS II and MDS-UPDRS III scores. Four patients in the probiotics group experienced mild reversible side effects.

## Conclusion

This study showed that consumption of a multi-strain probiotic (Hexbio®) over 8 weeks improved bowel opening frequency and whole gut transit time in PD patients with constipation.

## Introduction

Constipation is one of the commonest non-motor symptoms in Parkinson's disease (PD), reported in 80–90% of patients [1] and may precede the diagnosis of PD in 25% [2]. While the underlying pathogenesis of constipation in PD is complex, the enteric nervous system dysfunction due to alpha synuclein aggregation in the gut have been primarily implicated, leading to poor gastrointestinal motility, and outlet obstruction during defecation due to anal sphincter and puborectalis muscle dyssynergic contractions [1,3]. Recently, gut dysbiosis with alterations in faecal microbial composition was associated with the pathogenesis of PD and constipation [4,5] Effective and evidence-based treatment for constipation in PD up till recently was limited to iso-osmotic macrogol [6] and lubisprostone [7] which was shown to improve stool consistency and/or frequency. In line with gut dysbiosis, recent studies showed that probiotics therapy with or without prebiotics supplementation alleviated abdominal bloating and pain [8,9], improved stool consistency [8], improved stool frequency and the number of complete bowel movements [10]. Probiotics supplementation also improved motor severity scores, metabolic profiles namely hs-CRP levels, serum glutathione levels, and body-mass-index in PD patients, compared to placebo [11]. Published randomised trials [6,7,10] with positive outcomes on constipation in PD patients had evaluated stool frequency [10] and stool consistency [6] based on stool diary, but none so far had evaluated improvement in gut transit time as part of treatment objectives.

Here, we studied the efficacy of a multi-strain probiotic combined with a prebiotic fiber (fructo-oligosaccharide) compared to placebo, on constipation symptoms and intestinal motility, in PD patients with constipation.

## Methodology

### Study design

This was an eight-week investigator-initiated, double-blind, randomized, placebo-controlled intervention single center clinical trial involving 55 idiopathic PD patients attending the Parkinsons and neurology outpatient's clinic at Hospital Canselor Tuanku Muhriz, Kuala Lumpur, from October 2018 to February 2019. This study was conducted in accordance with the Declaration of Helsinki following approval by the institution's Ethics Committee for human

studies on the 6th October 2018. The Universiti Kebangsaan Malaysia Medical Research Committee gave an ethical approval to conduct this study with an approval code of FF-2018-387.

Written informed consent was obtained from all patients prior to enrolment. All data was analysed anonymously. Recruitment period was from the 15th October 2018 until 19th November 2018. The follow up period was from 15th October 2018 until 18th February 2019. Although registration of a clinical trial into a publicly listed clinical trials website is not a requirement by our institutions Ethics Committee, we attempted to register the study on the clinicaltrials.gov registry in April 2019. However, due to miscommunication regarding the primary account details, the registration was unfortunately overlooked and delayed, and was only successfully registered retrospectively in the clinicaltrials.gov registry on the 25th June 2020 with a clinical trial number (NCT04451096). The study conformed to the initial protocol, without any deviations in the methodology until study completion. The full study protocol is available as a supplementary file. The authors confirm that all ongoing and related trials for this drug/intervention are registered.

**Inclusion and exclusion criteria.** We included idiopathic PD patients [12] in Hoehn and Yahr stages 1–4, and fulfilled the Rome III criteria for functional constipation (Table 1).

Exclusion criteria included: MMSE score $\leq 21/30$; positive stool occult blood screening; diagnosis of secondary parkinsonism; previous history of small and large bowel disease; prior history of gastrointestinal tract surgery; use of probiotics or antibiotics two weeks prior to baseline, antidepressants or anticholinergics use; history of lactose intolerance; diagnosis of hypothyroidism and diabetes mellitus.

**Research objectives.** Our specific research objectives were whether probiotics could improve constipation symptoms and gut motility in PD patients with constipation. Additional research objectives included improvement in the quality of life and motor and non-motor symptoms of PD patients with probiotics.

**Sample size estimation.** The sample size was calculated to detect improvement in constipation symptoms using Power and Sample Size Calculation software [14], with two-sided 5% significance and 80% power based on a reference study by by Sakai et al [15] which assessed constipation symptoms in a healthy population who consumed fermented milk containing probiotic. The authors reported 40% and 10% changes in the produced hard or lumpy stools ($\geq 25\%$ of bowel movements) after 3 weeks of treatment among the treatment and control group, respectively. Additional 10% per group was added to compensate for drop-outs and non-response, with a final sample of 35 patients per group. A minimum of 25 patients per arm was selected for this study due to resource feasibility.

**Table 1. Rome III criteria for functional constipation adapted from Longstreth, et al [13].**

| | |
|---|---|
| Requires the presence of recurrent abdominal pain 3 days per month in the last 3 months, and symptom onset 6 months prior to diagnosis, with additional criteria below to be fulfilled as below: | |
| 1. Inclusion of two or more of the following: | |
| a. | Straining during at least 25% of defecations |
| b. | Lumpy or hard stools in at least 25% of defecations |
| c. | Sensation of incomplete evacuation for at least 25% of defecations d. Sensation |
| d | of anorectal obstruction/blockage for at least 25% of defecations |
| e. | Manual manoeuvres to facilitate at least 25% of defecations (e.g, digital evacuation, support of the pelvic floor) |
| f. | Fewer than 3 defecations per week |
| 2. Loose stools are rarely present without the use of laxatives | |
| 3. There are insufficient criteria for IBS | |

**Blinding, randomization and treatment assignment.** The study was conducted in a triple blinding manner. The patients, investigators, and the pharmacist distributing treatment were blinded to treatments assigned. Both study products (probiotic and placebo) were delivered in boxes with same study number by the sponsor (B-CROBES Laboratory Sdn. Bhd) and prelabelled as Treatment A and Treatment B. Each box contained identical sachets weighing 3 grams in sealed envelopes. Randomization and treatment assignment was performed by pharmacist (FT) using a computer-generated permuted block randomization method, in blocks of 4 which generated 6 different combination of sequence. Treatment allocation was assigned after enrolment and the assessor was blinded to the treatment assignment.

**Study drug and placebo composition.** The active treatment group received probiotic (Hexbio®) in orange flavouring containing microbial cell preparation of (MCP®BCMC®) at $30 \times 10^9$ colony forming units (CFU), 2% fructo-oliogosaccharide (FOS), and lactose. The microbial composition of the probiotics were: *Lactobacillus acidophilus* (BCMC® 12130)– 107mg, *Lactobacillus casei* (BCMC® 12313) -107mg, *Lactobacillus lactis* (BCMC® 12451)-107 mg, (BCMC® 02290) -107mg, *Bifidobacterium infantis* (BCMC® 02129) -107mg and *Bifidobacterium longum* (BCMC® 02120)-107mg. The placebo group received granulated milk of similar appearance to the probiotics containing lactose without fructo-oligosaccahride or microbial cells in orange flavouring. Both groups were instructed to consume one sachet mixed in a glass of water twice daily, before or after meals, for 8 weeks. Patients were contacted biweekly for adverse effects and compliance.

## Clinical assessments

Patients were assessed at baseline and at 8 weeks for all outcome assessments. Data on demographics, duration of PD, medication, H&Y Stage and the level of physical activity were recorded at baseline. Data on fiber intake history and physical exercise were extracted from the Garrigues Questionnaire.

## Primary outcome

**Constipation symptoms and Bowel Opening Frequency (Garrigues Questionnaire).** For the primary outcome on the presence of constipation and constipation symptoms, we used the he Garrigues Questionnaire (GQ). GQ is a 21-item self-report questionnaire which uses 2 sets of four-point Likert scale responses, and 'Yes' or 'NO' responses to evaluate the presence of constipation, constipation symptoms, fiber intake and the level of physical activity [16]. Of 21 items, 13 items assess bowel habits, while 9 items assess constipation symptoms specifically, including one item which asks for the number of bowel motion weekly (bowel opening frequency). The 9 items on constipation symptoms are: *(i)Feeling blockage in the anus;(ii) Need to press around anus/vagina to complete bowel movement; (iii) Spend >10 minutes to pass stool;(iv) Straining during bowel movement;(v) Feeling of hard stool;(vi) Feeling of incomplete emptying sensation; (vii)Bowel opening frequency (BOF);(viii) Frequency of oral laxative use; (ix) Frequency of enema use.*

The item on bowel opening frequency captured the absolute number of bowel movement (BM) per week. Patients were given a stool diary to record the weekly BM at baseline (two weeks prior to intervention) and 8 weeks (last week of intervention). The BM per week was additionally captured into a Likert scale as follows: < 3 per week, 3–5 per week and > 5 per week during analysis and included in our primary analysis together with the rest of the 8 GQ constipation items.

**Gut transit time (GTT).** The whole gut transit time (GTT) was measured using red carmine capsule, a non-absorbable, non-toxic colourant, which gives red colour to the stool [17].

Patients were asked to ingest four capsules in the morning on empty stomach. The time to have a red coloured stool from ingestion time was calculated as the GTT [17]. All subjects withheld enema/oral laxative one week prior to ingesting the capsules. The GTT was defined as delayed if ≥72 hours [18]. The GTT was compared at baseline and at 8 weeks between the groups.

**PD outcome assessments.** PD related assessments were performed during the ON period by single interviewer (AI). The MDS-UPDRS part II (Activities of Daily Living -13 items) and part III (Motor Assessment -18 items were used to determine the severity of PD. Each item was rated from 0 to 4. Higher scores indicate more severe disease. No adjustments to PD medications were made throughout the intervention.

Non-motor symptoms were assessed using the Non motor Symptom Scale (NMSS) consisting of 9 domains (cardiovascular, sleep/fatigue, mood/cognition, perceptual problems/hallucination, attention/memory, gastrointestinal tract, urinary, sexual function and miscellaneous). Frequency and severity of each item were multiplied and summed to give a total score. Higher scores indicate more disability.

Quality of life was measured using the Parkinson's disease Questionnaire 39 summary indices (PDQ39-SI). The score was calculated by dividing the sum of the total raw score by the maximum possible score 156 or 152 points and multiplying by 100.

## Sub analysis

The following sub analyses were not part of the initial study design and were conducted after study completion.

**Garrigues Questionnaire (BOF) item.** The absolute number of BM per week was categorised into: < 3 per week, 3–5 per week and > 5 per week during analysis and tabulated together with the rest of the 8 GQ constipation items.

**Frequency of patients with constipation.** From the stool diary, the percentage of patients who experienced < 3 BM per week was calculated and between-group frequencies were compared at baseline and at 8 weeks.

**Frequency of patients with delayed GTT.** The percentage of patients with delayed GTT and the mean change in the GTT (GTT 8 weeks—GTT baseline) were compared between the two groups at baseline and at 8 weeks.

**Body-mass-index.** Height and weight of subjects were determined using a standard scale. Body-mass-index (BMI) was calculated by dividing height in meters squared to weight in kilogram. Baseline BMI and change in BMI between-groups and within-group were compared.

## Statistical analysis

Data obtained was analysed using SPSS version 19. Continuous data was described as mean with standard deviation (SD) or median with interquartile range (IQR) depending on normality of data. Qualitative data was expressed as frequencies (*n*) and percentages (%). For comparison of normally and non-normally distributed continuous data, *t*-test and Mann–Whitney *U* test were used, respectively for between-groups differences. For both primary and secondary outcomes, within-groups differences continuous data were analysed using either paired t-test or Wilcoxon Signed rank test. For the subanalysis comparison of proportions between two groups, Pearson chi square, Fisher's Exact, Continuity correction tests were used following an intention-to-treat analysis. Treatment effect and its respective 95% confidence interval (CI) at 8 weeks of treatment was assessed using various regression models under the generalized linear models. A corrected *p* value of <0.005 was regarded as statistically significant for the constipation outcome. The uncorrected *p* value <0.05 was reported for the gut motility and secondary

PD outcomes. Analyses were done on an intention-to-treat basis, including all eligible patients who were randomised and who returned for a week-8 visit. All analysis was performed while investigators were still blinded. The sponsor was not involved in data acquisition or analysis.

# Results

## Baseline demographics

Of seventy patients screened, fifty-five were recruited: 27 patients received probiotic and 28 received placebo. The first patient recruitment began on the 9th November 2018 and last patient follow-up was on the 14th February 2019. In the first week of recruitment into the study, 4 patients allocated to probiotic therapy dropped out due to side effects of abdominal bloating ($n = 2$) and dizziness ($n = 2$). Another patient in the probiotic group was lost to follow up at week 1. Two patients in the placebo group declined participation in the first week of recruitment. The study flow is shown in Fig 1.

The baseline characteristics including median age, duration of PD, Hoehn & Yahr stage, BMI, medication use, education level, level of physical activity and fibre intake were comparable for both groups (Table 2).

## Primary outcome

**Constipation symptom and bowel opening frequency (Garrigues Questionnaire).**
There was no baseline difference was observed in the in self-reported constipation symptoms.

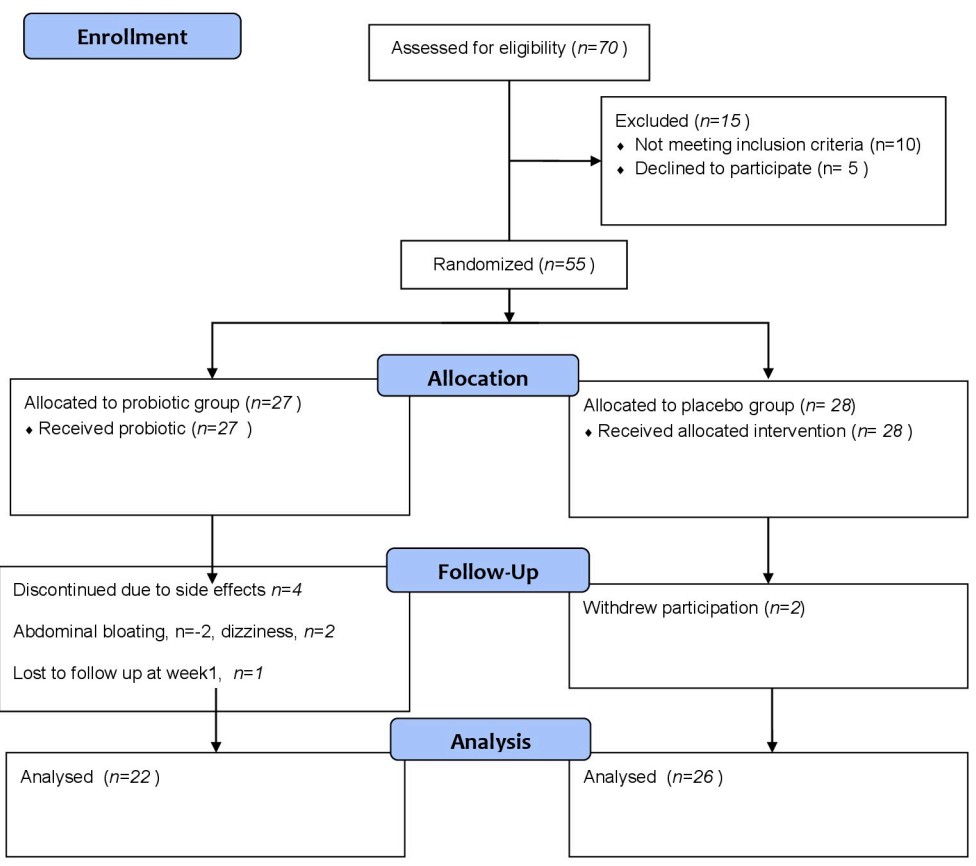

**Fig 1. Study flow diagram.**

**Table 2. Between-group comparison of baseline demographics, clinical parameters and life style factors.**

|  | Probiotics | Placebo | Statistical test | *p* value |
|---|---|---|---|---|
|  | *n* = 27 | *n* = 28 |  |  |
| **Baseline demographics** |  |  |  |  |
| Age, years (median (IQR)] | 69.0 (64.0–74.0) | 70.5 (62.0–70.3) | -0.23[a] | 0.820 |
| *Sex, n (%)* |  |  |  |  |
| Male | 16 (59.3) | 17 (60.7) | 0.12[b] | 0.912 |
| Female | 9 (40.9) | 10 (38.5) |  |  |
| Duration of illness (median, IQR) | 6.0 (5.0–10.0) | 6.5 (3.3–11.5) | -0.20[a] | 0.846 |
| *Race, n (%)* |  |  |  |  |
| Malay | 15 (55.6) | 11 (39.3) | 2.09[c] | 0.346 |
| Chinese | 12 (44.4) | 16 (57.1) |  |  |
| Indian | 0 | 1 (3.6) |  |  |
| Sedentary lifestyle (Physical exercise < 4 hour per week), n (%) | 14 (51.9) | 13 (48.1) | 0.16[b] | 0.898 |
| Levodopa use, n (%) | 25(92.6) | 25 (89.3) | 0[d] | 1 |
| Dopamine agonist use, n (%) | 17 (63) | 16 (57.1) | 0.19[b] | 0.660 |
| *Fibre intake, n (%)* |  |  | 1.744[c] | 0.503 |
| Low (less than 3 serving per day) | 22 (81.5) | 21 (75) |  |  |
| Medium (3–5 serving per day) | 4 (14.8) | 7 (25) |  |  |
| *PD Stage (Hoehn & Yahr), n (%)* |  |  | 1.15[b] | 0.701 |
| ≤ 3 | 16 (59.3) | 18 (64.3) |  |  |
| > 3 | 11 (40.7) | 10 (35.7) |  |  |

[a]Mann-Whitney;
[b]Pearson chi-square;
[c]Fisher's Exact test;
[d]Continuity correction.

The mean weekly BM was comparable between both groups [probiotic: 2.07 (SD 0.73) vs placebo: 1.96 (SD 0.33); *p* = 0.473] (Table 3).

At 8 weeks, the mean weekly BM was significantly higher in the probiotic group [4.18 (SD1.44)] compared to placebo [2.81 (SD1.05)] (mean difference 1.37, 95% CI 0.68, 2.07, uncorrected *p*<0.001). Patients in the probiotic group reported five times higher odds (odds ratio = 5.48, 95% CI 1.57, 19.12, uncorrected *p* = 0.008) for increasing BOF (< 3 to 3–5 to >5 times/week) compared to the placebo group. There were no significant differences in the other 8 items between the two groups at 8 weeks, including on enema use (Table 4).

**Gut transit time (GTT).** At baseline, the mean GTT was similarly prolonged in both groups (Table 3). At 8 weeks, the mean GTT reduced significantly in the probiotic group [77.32 (SD 55.34) hours]compared to placebo [113.54 (SD 61.53) hours] (mean difference -36.22, 95% CI -68.90, -3.54, *p* = 0.030). The mean change in GTT from baseline was more significant in the probiotic group [58.05 (SD 59.30) hours] compared to the placebo [20.73 (SD 60.48) hours] (mean difference 37.32, 95% CI 4.00, 70.63, *p* = 0.028) (Table 4).

## Secondary outcome

**PD related assessments.** The MDS-UPDRS II and III scores, NMSS scores and PDQ-39 SI scores were comparable at baseline (Table 3). At 8 weeks, no significant differences were observed in the MDS-UPDRS 11 and III scores, NMSS scores and PDQ-39SI scores, between the two groups (Table 4).

**Table 3. Between-group comparison of primary constipation outcome, gut motility, secondary PD outcomes and BMI at baseline.**

| | Probiotics | Placebo | Statistical test | p value |
|---|---|---|---|---|
| **Primary Outcome** | *n* = 27 | *n* = 28 | | |
| **Garrigues Questionnaire Items** | | | | |
| *Feel hard stools, n (%)* | | | 2.41[a] | 0.300 |
| Never | 0 | 0 | | |
| Sometimes (<25%) | 4 (14.8) | 9 (32.1) | | |
| Often (≥25%) | 18 (66.7) | 14 (50) | | |
| Always | 5 (18.5) | 5 (17.9) | | |
| *Incomplete sensation, n (%)* | | | 1.89[b] | 0.608 |
| Never | 4 (14.8) | 2 (7.1) | | |
| Sometimes (<25%) | 8 (29.6) | 9 (32.1) | | |
| Often (≥25%) | 10 (37) | 14 (50) | | |
| Always | 5 (18.5) | 3 (10.7) | | |
| *Feel blockage in anus, n (%)* | | | 1.86[b] | 0.613 |
| Never | 7 (25.9) | 11 (39.3) | | |
| Sometimes (<25%) | 4 (14.8) | 5 (17.9) | | |
| Often (≥25%) | 12 (44.4) | 10 (35.7) | | |
| Always | 4 (14.8) | 2 (7.1) | | |
| *Need to press around anus/vagina, n (%)* | | | 4.81[b] | 0.172 |
| Never | 18 (66.7) | 15 (53.6) | | |
| Sometimes (<25%) | 4 (14.8) | 10 (35.7) | | |
| Often (≥25%) | 4 (14.8) | 1 (3.6) | | |
| Always | 1 (3.7) | 2 (7.1) | | |
| *Spend >10 minutes to pass stool, n (%)* | | | 5.96[b] | 0.086 |
| Never | 0 | 2 (7.1) | | |
| Sometimes (<25%) | 1 (3.7) | 6 (21.4) | | |
| Often (≥25%) | 17 (63) | 12 (42.9) | | |
| Always | 9 (33.3) | 8 (28.6) | | |
| *\*Bowel opening frequency, n (%)* | | | 1.24[b] | 0.741 |
| Never | 0 | 0 | | |
| < 3 times /week | 25 (92.6) | 27 (96.4) | | |
| 3–5 times /week | 1 (3.7) | 1 (3.6) | | |
| > 5 times /week | 1 (3.7) | 0 | | |
| BM /week [mean (SD)] | 2.07 (0.73) | 1.96 (0.33) | 0.72[c] | 0.473 |
| *Frequency of enema use, n (%)* | | | 5.13[b] | 0.159 |
| Never | 18 (66.7) | 17 (60.7) | | |
| Less than 1/week | 1 (3.7) | 4 (14.3) | | |
| 1 or more /week | 3 (11.1) | 6 (21.4) | | |
| Everyday | 5 (18.5) | 1 (3.6) | | |
| *Strain during bowel movement, n (%)* | | | 1.92[a] | 0.589 |
| Never | 2 (7.4) | 0 | | |
| Sometimes (<25%) | 3 (11.1) | 3 (10.7) | | |
| Often (≥25%) | 14 (51.9) | 17 (60.7) | | |
| Always | 8 (29.6) | 8 (28.6) | | |
| *Frequency of oral laxative use, n (%)* | | | 5.12[a] | 0.163 |
| Never | 8 (29.6) | 10 (35.7) | | |
| Sometimes (<25%) | 1 (3.7) | 2 (7.1) | | |
| Often (≥25%) | 7 (25.9) | 12 (42.9) | | |

*(Continued)*

**Table 3.** (Continued)

| | Probiotics | Placebo | Statistical test | p value |
|---|---|---|---|---|
| **Primary Outcome** | *n* = 27 | *n* = 28 | | |
| Always | 11 (40.7) | 4 (14.3) | | |
| **Gut Motility Outcome** | | | | |
| GTT [mean (SD)] | 125.26 (54.81) | 128.46 (53.68) | -0.22[c] | 0.827 |
| Delayed GTT (≥72 hours), n (%) | 19 (70.4) | 23 (82.1) | 1.05[a] | 0.304 |
| BM < 3 per week, n (%) | 25 (92.6) | 27 (96.4) | $1.05 \times 10^{-3}$[d] | 0.974 |
| **PD related Outcome** | | | | |
| NMSS [median (IQR)] | 68.0 (39.0–84.0) | 71.0 (44.8–99.0) | -0.68[e] | 0.495 |
| PDQ39-SI [median (IQR)] | 30.8 (17.9–50.6) | 36.7 (23.8–59.8) | -1.07[e] | 0.285 |
| UPDRS Part II [median (IQR)] | 16.0 (12.0–24.0) | 18.0 (11.5–28.0) | -0.71[e] | 0.479 |
| UPDRS Part III [median (IQR)] | 29.0 (20.0–53.0) | 27.5 (20.3–45.8) | 0[e] | 1 |
| **BMI [mean (SD)]** | 22.94 (4.65) | 22.84 (5.16) | 0.08[c] | 0.941 |

[a]Fisher's Exact test;

[b]*t*-test;

[c]Pearson chi-square;

[d]Continuity correction;

[e]Mann-Whitney.

UPDRS = Unified Parkinson's Disease Rating Scale, NMSS = Non-Motor Symptoms Score, PDQ 39 SI- Parkinson's disease Questionnaire, GCQ = Garrigues

Constipation Questionnaire, BM = bowel motion, GTT = gut transit time;

*GCQ item as ordinal data.

## Subanalysis

**Frequency of patients with constipation (BM< 3 per week).** At baseline, 25 (92.6%) patients in the placebo group and 27 (96.4%) patients in the probiotic group had < 3 BM per week (*p* = 0.974). At 8 weeks, only 5 (22.7%) patients in the probiotic group had < 3 BM per week compared to 15 (57.7%) patients in the placebo group, Thus, patients in the probiotic group had low odds for < 3 BM per week (odds ratio = 0.22, 95% CI 0.06, 0.76, *p* = 0.017) compared to placebo group (Table 4).

**Frequency of patients with delayed GTT.** The proportion of patients with delayed GTT at baseline were similar in probiotic (n = 19;70.4%) and placebo group (n = 23; 82.1%) (*p* = 0.304) (Table 2). At 8 weeks, only 8 (36.4%) patients in the probiotic group had delayed GTT compared to 16 (61.5%) patients in the placebo group (*p* = 0.086) (Table 4).

**Body mass index.** There were no significant differences in the BMI between the probiotic and placebo group, at 8 weeks (Table 4).

**Within-group analysis for GTT, PD outcomes, and BMI.** Within-group analysis showed significant improvement in the GTT at 8 weeks, in the probiotic group from 125.26 (SD54.81) hours to 77.32(SD55.35) hours, *p* <0.001. No significant difference was observed in the placebo group. The MDS-UPDRS II (MDS UPDRS III median NMSS and median PDQ-39SI scores in the probiotic group significantly improved compared to baseline (Table 5). For the placebo group, there was significant improvement in the NMSS scores (*p* = 0.007) compared to baseline, but no significant improvements were observed in the PDQ-39S, MDS UPDRS II and III (Table 5). There was significant improvement in BMI in the probiotic group from 22.0 to 22.9 kg/m² (*p* = 0.010) (Table 5).

**Table 4. Between-group comparison of primary constipation outcome, gut motility, secondary PD outcomes and BMI at 8 weeks.**

| | Probiotic | Placebo | Statistical test | p value | Treatment effect (95% CI) |
|---|---|---|---|---|---|
| **Primary Outcome** | **n = 22** | **n = 26** | | | |
| **Garrigues Questionnaire** | | | | | |
| *Feel Hard stools, n (%)* | | | 1.15 | 0.284 | 0.54 (0.18, 1.66) |
| Never | 3(13.6) | 4 (15.4) | | | |
| Sometimes (<25%) | 15 (68.2) | 12(46.2) | | | |
| Often (≥25%) | 4 (18.2) | 10 (38.5) | | | |
| Always | 0 (0) | 0 (0) | | | |
| *Incomplete sensation, n (%)* | | | 1.27 | 0.26 | 0.54 (0.19, 1.57) |
| Never | 11(50) | 10 (38.5) | | | |
| Sometimes (<25%) | 8(36.4) | 8(30.8) | | | |
| Often (≥25%) | 2 (9.1) | 7 (26.9) | | | |
| Always | 1 (4.5) | 1 (3.8) | | | |
| *Feel blockage in anus, n (%)* | | | 0.58 | 0.448 | 1.52 (0.51,4.51) |
| Never | 10(45.5) | 15(57.7) | | | |
| Sometimes (<25%) | 8(36.4)) | 7(26.9) | | | |
| Often (≥25%) | 4(18.2) | 4(15.4) | | | |
| Always | 0 (0) | 0(0) | | | |
| *Need to press around anus/vagina, n (%)* | | | 0.10 | 0.756 | 0.80 (0.20, 3.27) |
| Never | 18 (81.8) | 20 (76.9) | | | |
| Sometimes (<25%) | 2 (9.1) | 5 (19.2) | | | |
| Often (≥25%) | 1 (4.5) | 0 (0) | | | |
| Always | 1 (4.5) | 1 (3.8) | | | |
| *Spend >10 minutes to pass stool, n (%)* | | | 0.09 | 0.768 | 0.86 (0.31, 2.40) |
| Never | 5(22.7) | 6(23.1) | | | |
| Sometimes (<25%) | 8(36.4) | 7(26.9) | | | |
| Often (≥25%) | 7(27.3) | 10(38.5) | | | |
| Always | 3(13.6) | 2(7.7) | | | |
| *[*]Bowel opening frequency, n (%)* | | | 7.12 | 0.008 | 5.48 (1.57, 19.12) |
| < 3 times /week | 5(22.7) | 15(57.7) | | | |
| 3–5 times /week | 14 (63.6) | 11 (42.3) | | | |
| > 5 times /week | 3 (13.6) | 0 (0) | | | |
| BM /week [mean (SD)] | 4.18 (1.44) | 2.81 (1.06) | - | <0.001* | 1.37 (0.68, 2.07) |
| *Frequency of enema use, n (%)* | | | 0.14 | 0.712 | 0.77 (0.19, 3.14) |
| Never | 18 (81.8) | 20(76.9) | | | |
| Less than 1/week | 1(4.5) | 1(3.8) | | | |
| 1 or more /week | 2(9.1) | 5(19.2) | | | |
| Everyday | 1(4.5) | 0 | | | |
| *Strain during bowel movement, n (%)* | | | 3.06 | 0.080 | 0.37 (0.13, 1.13) |
| Never | 6 (27.3) | 5 (19.2) | | | |
| Sometimes (<25%) | 13 (59.1) | 10 (38.5) | | | |
| Often (≥25%) | 2 (9.1) | 8 (30.8) | | | |
| Always | 1(4.5) | 3 (11.5) | | | |
| *Frequency of oral laxative use, n (%)* | | | 0.11 | 0.744 | 1.21 (0.39, 3.76) |
| Never | 13(59.1) | 17(65.4) | | | |
| Sometimes (<25%) | 1 (4.5) | 2 (7.7) | | | |
| Often (≥25%) | 6 (27.3) | 3 (11.5) | | | |
| Always | 2 (9.1) | 4 (15.4) | | | |

*(Continued)*

**Table 4.** (Continued)

| | Probiotic | Placebo | Statistical test | p value | Treatment effect (95% CI) |
|---|---|---|---|---|---|
| **Primary Outcome** | **n = 22** | **n = 26** | | | |
| **Gut Motility Outcome** | | | | | |
| GTT, hours [mean (SD)] | 77.32 (55.35) | 113.54 (61.54) | - | 0.030 | -36.22 (-68.90, -3.54) |
| Mean difference in GTT from baseline, hours [mean (SD)] | 58.04 (59.04) | 20.73 (60.48) | - | 0.028 | 37.32 (4.00, 70.63) |
| Delayed GTT ≥72 hours, n (%) | 8 (36.4) | 16 (61.5) | 2.96 | 0.086 | 0.36 (0.11, 1.16) |
| BM < 3 times /week, n (%) | 5 (22.7) | 15 (57.7) | 5.65 | 0.017 | 0.22 (0.06, 0.76) |
| **PD outcome** | | | | | |
| NMSS [median (IQR)] | 28.2 (9.5–42.9) | 34.8 (23.4–50.3) | - | 0.054 | -19.82 (—40.01, 0.37) |
| PDQ39-SI [median (IQR)] | 50.0 (27.0–65.5) | 63.0 (35.0–92.5) | - | 0.136 | -9.11 (-21.10, 2.88) |
| UPDRS Part II [median (IQR)] | 15.0 (10.8–20.8) | 17.0 (12.3–29.3) | - | 0.172 | -3.90 (-9.50, 1.69) |
| UPDRS Part III [median (IQR)] | 19.0 (12.8–36.0) | 30.5 (15.8–46.0) | - | 0.103 | -10.23 (-22.53, 2.06) |
| BMI [mean(SD)] | 23.17 (4.80) | 22.58 (5.11) | - | 0.673 | 0.59 (-2.17, 3.36) |

Primary outcome: Significant at $p$ <0.005; Gut Motility and PD outcomes: Significant at $p$ <0.05.

[a]generalized linear models (ordinal logistic, binary logistic, linear regressions) (difference between the probiotic and placebo groups).

UPDRS = Unified Parkinson's Disease Rating Scale, NMSS = Non-Motor Symptoms Score, PDQ 39 SI- Parkinson's disease Questionnaire, GCQ = Garrigues

Constipation Questionnaire, BM = bowel motion, GTT = whole gut transit time;

*GCQ item as ordinal data; CI: Confidence interval.

**Side effects.** Four of 27 patients (14.8%) in the probiotic group experienced side effects of abdominal bloating ($n = 2$) and dizziness ($n = 2$) and dropped out at week 1 These side effects were transient and resolved with discontinuation of the probiotic. No side effects occurred in the remaining 22 patients in the probiotic group throughout the study period. There were no adverse events recorded in the placebo group.

**Table 5. Within-group comparison of GTT, PD outcome parameters and BMI at 8 weeks.**

| Probiotic | Gut motility[a] | Baseline, mean (SD) | 8 week, mean (SD) | Statistical test | p value |
|---|---|---|---|---|---|
| | GTT, hours | 125.26 (54.81) | 77.32 (55.35) | 4.59 | <0.001 |
| | **PD Parameters[b]** | **Baseline, median (IQR)** | **8 week, median (IQR)** | | |
| | PDQ39- SI | 30.8 (17.9–50.6) | 28.2 (9.5–42.9) | -2.48 | 0.013 |
| | NMSS | 68.0 (39.0–84.0) | 50.0 (27.0–65.5) | -3.87 | <0.001 |
| | UPDRS Part 11 | 16.0 (12.0–24.0) | 15.0 (10.8–20.8) | -2.05 | 0.040 |
| | UPDRS Part 111 | 29.0 (20.0–53.0) | 19.0 (12.8–36.0) | -3.85 | <0.001 |
| | BMI | 22.0 (20.0–25.0) | 22.9 (19.8–26.3) | -2.56 | 0.010 |
| Placebo | Gut motility[a] | Baseline, mean (SD) | 8 week, mean (SD) | | |
| | GTT, hours [mean (SD)] | 128.46 (53.68) | 113 (61.54) | 1.75 | 0.093 |
| | **PD Parameters[b]** | **Baseline, median (IQR)** | **8 week, median (IQR)** | | |
| | PDQ39- SI | 36.7 (23.8–59.8) | 34.8 (23.4–50.3) | -0.95 | 0.341 |
| | NMSS | 71.0 (44.8–99.0) | 63.0 (35.0–92.5) | -2.69 | 0.007 |
| | UPDRS Part 11 | 18.0 (11.5–28.0) | 17.0 (12.3–29.3) | -0.48 | 0.634 |
| | UPDRS Part 111 | 27.5 (20.3–45.8) | 30.5 (15.8–46.0) | -1.43 | 0.154 |
| | BMI | 22.8 (19.4–26.9) | 23.0 (18.2–27.0) | -0.60 | 0.552 |

[a]GTT outcome parameters: Paired $t$-test;

[b]PD outcome parameters: Wilcoxon Signed Rank test.

BM = bowel motion; GTT = gut transit time; UPDRS = Unified Parkinson's Disease Rating Scale; NMSS = Non-Motor Symptoms Score; PDQ 39 SI- Parkinson's disease Questionnaire.

## Discussion

Constipation is a prevalent non-motor symptom in PD. Its underlying pathophysiology is complex with an interplay of various contributory factors, such as disease-related gastrointestinal dysfunction due to alpha-synuclein accumulation within the enteric nervous system [1,3], side effects of anti-parkinson medications [19], life style risk factors and superimposed physical weakness due to frailty [20,21], cumulatively resulting in slowed intestinal motility, and dyssynergic anal sphincter contractions.

This study showed that the consumption of a multi-strain probiotic (Hexbio®) over 8 weeks, significantly improved bowel opening frequency and gut transit time in PD patients with constipation. PD patients who consumed probiotics experienced a significantly higher mean weekly bowel movement (4.18) compared to the placebo group (2.81). Patients who received probiotics reported increased weekly BOF, with 13.6% experiencing more than 5 BM per week. Additionally, the percentage of patients who remained constipated (< 3 BM per week) was also significantly lower in the probiotic group (22.7%) compared to 57.7% in the placebo group. The item assessing BOF was identified as one of the most discriminant items of GQ for detecting constipation in the general population [16]. No changes were reported in the other items of GQ, including two other discriminant items which evaluated hard stools and defecation difficulties [16]. The lack of improvement in the other items could be partly contributed by the fact that GQ is an instrument for the detection of constipation, rather than the severity of constipation [22,23]. As there were no PD-specific constipation assessment questionnaires at the time this study was conducted, we used the GQ, as it had incorporated the ROME III criteria in its screening items [16,22]. A study comparing different constipation tools showed that GQ had the highest number of items which were congruent with the ROME III criteria, compared to other constipation assessment tools [16,22,23]. However, we acknowledge that while being sensitive for the diagnosis of constipation, the changes in severity of constipation are not adequately assessed by GQ. Retrospectively, it seemed that the item on BOF may have offered the most objective evaluation of self-reported constipation in this study, as it measured the actual number of weekly bowel movements compared to the other items of GQ, which used a Likert scale [16]. Of noteworthy, at baseline, a significantly higher percentage of patients in the probiotic group had required frequent enema. At 8 weeks, this need for enema was no different to the placebo group suggesting that probiotics treatment had resulted in less requirement for enema in this group.

Probiotic therapy is well studied in non-PD adults with functional constipation. A recent meta-analysis concluded that multiple strain probiotics were effective in improving stool frequency, whole gut transit time and stool consistency patients with functional constipation [24]. still limited to a few studies, there are accumulating data supporting the benefits of probiotics in PD patients with constipation [8–10,25] and without constipation [11]. However, these studies mainly evaluated self-reported constipation symptoms [8,9] and the absolute number of bowel movements [10], but did not evaluate intestinal motility. The first study to establish beneficial effects of probiotics in PD patients with constipation showed that daily fermented milk containing *Lactobacillis casei Shirota* strain over 5 weeks significantly improved stool consistency, reduced abdominal bloating and pain and sensation of incomplete emptying, when compared to dietetic therapy alone [8]. Another trial which evaluated probiotics with a prebiotic fiber supplementation in PD patients with constipation over 4 weeks, showed that probiotics improved frequency of complete bowel motions compared to placebo [10]. More recently another study showed that multi-strain probiotics improved spontaneous bowel movement, and quality of life scores associated with constipation compared to placebo over 4 weeks in PD patients with constipation [26]. Probiotics supplementation over 3 months

alleviated abdominal pain and bloating supplementation compared to trimebutine 200mg thrice daily in PD patients with constipation [9]. The findings of our study thus strengthen the positive findings on probiotics in PD-related constipation.

The most notable finding from the present study, which paralleled the improvement in BOF, was on the GTT. The mean baseline GTT was prolonged ($\geq$ 72 hours) in both groups. With probiotic therapy, the GTT reduced significantly from 135 hours at baseline to 77 hours, with an overall reduction of 58.05 hours, whereas in the placebo group, there was a slight and non-significant reduction in the WGTT from 134 hours to 114 hours (-20.73 hour). To date, no studies had directly evaluated the effects of probiotics on GTT in PD patients. Taken together, our findings suggest that multi-strain probiotic (Hexbio®) supplementation over 8 weeks improved bowel opening frequency in PD patients, most likely due to improved gastro-intestinal motility.

The improvement in bowel opening frequency and gut motility with probiotics in this study may be attributed to its beneficial effect on gut dysbiosis. Gut dysbiosis was shown to precede the development of PD and has been implicated in the pathogenesis of functional constipation, especially the slow transit type. Previous studies showed that PD patients have altered composition of gut microbiota [27–29]. Moreover, faecal samples of PD patients demonstrated differing microbiota composition and lower concentration of short chain fatty acids (SCFAs) compared to controls [27,28]. Physiological levels of SCFAs are crucial to gut health. SCFAs have anti-inflammatory properties and are important for gut mucosal lining repair [27], in modulating the activity of enteric nervous system, thereby enhancing gut motility [30]. SCFAs are produced by the fermentation of complex dietary carbohydrates (prebiotics) by colonic bacteria in the gut. Adequate amounts of undigestible dietary fibers (prebiotics) and colonic bacterial composition are required to ensure adequate levels of SCFA in the gut. SCFAs reduce luminal pH, enhance colonic peristalsis, and shorten whole gut transit time (GTT) [31,32]. A short-term randomised clinical study among Malaysian adults with functional constipation who consumed the same probiotic-prebiotic combination (Hexbio®) as used in our study, reported an improvement in bowel opening frequency and alleviation of constipation symptoms after 7 days [33]. We thus hypothesize that the synergistic effects of probiotic with a prebiotic (FOS) may have acted as a 'synbiotic' [34], in improving intestinal motility and bowel opening frequency, possibly by facilitating the production of SCFA from as early as 7 days. This, however, needs to be confirmed with further studies evaluating gut microbiota composition and stool analysis for SCFA.

Although, no significant differences were observed in any of the PD-related outcomes between the groups, within-group analysis showed significant improvement in the BMI, MDS-UPDRS part II and III scores, NMSS scores and PDQ-39SI scores in the probiotic group. Notably, the BMI improved significantly from 22.0 kg/m$^2$ to 22.9 kg/m$^2$, which is rather exciting, considering PD patients are prone to weight loss and sarcopenia. We postulate that this increase in BMI may be attributed to improved gut enterocyte function with probiotics, leading to better absorption of nutrients. A randomised controlled study showed that probiotics supplementation over 12 weeks led to improved BMI and motor severity scores (MDS-UPDRS), and a reduction in metabolic and inflammatory parameters, such as highly sensitive C reactive protein, serum glutathione and serum insulin levels in PD patients [11].

Probiotic was well tolerated throughout the study, except for four patients who withdrew in the first week due to abdominal bloating and dizziness. These side effects resolved fully following discontinuation of probiotics. There were no side effects in the remaining patients who received probiotics throughout the study.

We acknowledge that there are limitations to our study. Firstly, we had used GQ which was developed to detect the presence of constipation-related symptoms, rather than constipation severity, and hence may not have captured improvement in constipation severity accurately.

Additionally, a Bristol stool chart would have been able to give a more objective assessment of stool form. Secondly, while the updated ROME IV may have facilitated an earlier diagnosis of constipation, we believe the ROME III criteria used in our inclusion criteria was adequate to detect constipation in our patients, as constipation in PD is typically chronic and may even predate the motor symptoms. Hence we believe the use of ROME III would not have led to significant false negative or bias in the recruitment of patients [35]. Thirdly, despite randomization, both groups differed in their need for enema. A higher need for enema in the probiotics group indicated that they were perhaps more severely constipated. Fourthly, due to possibility of recall bias, we did not perform a detailed dietary profile to determine the total fibre intake, total amount of food and water consumption, we are unable to account for the possible contributions of other diet components, such as fatty acids, phytochemicals, and vitamins, which could serve as bacterial substrate for metabolites formation. However, patients were asked to follow their usual dietary intake throughout the study. Additionally, we were unable to correlate the improvement in bowel opening frequency with changes in the composition of host microbiota and the levels of SCFA as we did not perform a quantitative analysis of short chain fatty acid concentration (SCFA) or faecal microbiota composition. Finally, we could only perform intention-to-treat analysis (ITT) for one of the study outcomes (difference in WGTT). For the rest of the study outcomes, ITT analysis could not be performed as patients in both groups dropped out within the first week of trial and were not keen for further follow up.

Despite these limitations, we believe our study has a number of strengths. Firstly, this was a randomized controlled trial which adhered to strict blinding methods. Patients and all investigators were only un-blinded after analysis. Secondly, in addition to self-reports of constipation symptoms, we also evaluated the effects of probiotics on gut motility (GTT), which had not been evaluated by previous studies on PD-related constipation. Thirdly, all clinical assessments particularly the MDS-UPDRS and NMSS were performed by a single investigator to reduce interrater variability. Additionally, despite a relatively small sample size, we were able to detect significant changes between groups with good treatment effect for both primary and secondary outcome parameters. However, these significant changes must be interpreted with caution, due to the small sample size.

## Generalisability

It is perhaps reasonable to infer that the probiotic used in this study would be effective in all PD patients with constipation. However, as there are different types of probiotics and combination of probiotics available, it would be difficult to ascertain if all probiotics would confer the same benefit. As our patients were from a multi-ethnic background, with differing dietary practices, studies in other population maybe required to validate our findings.

## Conclusions

This study showed that Hexbio® containing MCP®BCMC® strains was safe and effective in improving bowel opening frequency and gastrointestinal motility in PD patients with constipation. While, these findings strengthen existing evidence on the efficacy of probiotics on constipation in PD, the clinical benefits observed should be interpreted with caution due to limitations in achieving the required sample size. Multi-centre studies with larger sample size are required to validate our findings and if the positive effects could be generalised to other populations with differing dietary background. Future research should also focus on evaluating long term effects of probiotics on sustaining positive changes in gut microbiota composition beyond intervention duration.

## Supporting information

**S1 Checklist. CONSORT 2010 checklist of information to include when reporting a rando-mised trial**\*.
(DOC)

**S1 File.**
(DOCX)

## Author Contributions

**Conceptualization:** Azliza Ibrahim, Raja Affendi Raja Ali, Mohd Rizal Abdul Manaf, Farah Waheeda Tajurruddin, Siti Hajar Md Desa, Norlinah Mohamed Ibrahim.

**Data curation:** Norfazilah Ahmad, Norlinah Mohamed Ibrahim.

**Formal analysis:** Mohd Rizal Abdul Manaf, Norfazilah Ahmad, Wong Zhi Qin.

**Funding acquisition:** Raja Affendi Raja Ali.

**Investigation:** Azliza Ibrahim, Wong Zhi Qin, Norlinah Mohamed Ibrahim.

**Methodology:** Azliza Ibrahim, Raja Affendi Raja Ali, Mohd Rizal Abdul Manaf, Norfazilah Ahmad, Farah Waheeda Tajurruddin, Wong Zhi Qin, Siti Hajar Md Desa, Norlinah Mohamed Ibrahim.

**Project administration:** Siti Hajar Md Desa, Norlinah Mohamed Ibrahim.

**Supervision:** Raja Affendi Raja Ali, Mohd Rizal Abdul Manaf, Farah Waheeda Tajurruddin, Wong Zhi Qin, Siti Hajar Md Desa, Norlinah Mohamed Ibrahim.

**Validation:** Mohd Rizal Abdul Manaf, Norlinah Mohamed Ibrahim.

**Writing – original draft:** Azliza Ibrahim.

**Writing – review & editing:** Raja Affendi Raja Ali, Mohd Rizal Abdul Manaf, Norfazilah Ahmad, Farah Waheeda Tajurruddin, Wong Zhi Qin, Siti Hajar Md Desa, Norlinah Mohamed Ibrahim.

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
