## [Decision Letter · Decision Letter 0]

8 Oct 2020

PONE-D-20-21349

Probiotic-prebiotic therapy improved constipation and gut motility in Parkinson’s disease: A Randomised controlled trial

PLOS ONE

Dear Dr. Ibrahim,

Thank you for submitting your manuscript to PLOS ONE. After careful consideration, we feel that it has merit but does not fully meet PLOS ONE’s publication criteria as it currently stands. Therefore, we invite you to submit a revised version of the manuscript that addresses the points raised during the review process.

The manuscript has been evaluated by four reviewers, and their comments are available below. You will see the reviewers have commended you for addressing this underexamined theme in the literature. However, they have also raised a number of concerns that should be addressed before the manuscript can be further considered for publication.

The key concerns noted by the reviewers relate to the limitations of the study sample and statistical analyses. In particular, Reviewer 3 noted the need for sensitivity analyses due to the high dropout rate, while Reviewer 1 highlighted a need for clarification about the sample size calculations. Furthermore, the reviewers raised concerns about multiple testing, via the multiple t-test analyses. These limitations have implications for the interpretation of the results.

Additionally, the reviewers requested more information about the name of the probiotic strains used in this study.

We look forward to receiving your revised manuscript.

Kind regards,

Danielle Poole

Staff Editor

PLOS ONE

Journal Requirements:

2. Thank you for submitting your clinical trial to PLOS ONE and for providing the name of the registry and the registration number. The information in the registry entry suggests that your trial was registered after patient recruitment began. PLOS ONE strongly encourages authors to register all trials before recruiting the first participant in a study.

1) your reasons for your delay in registering this study (after enrolment of participants started);

2) confirmation that all related trials are registered by stating: “The authors confirm that all ongoing and related trials for this drug/intervention are registered”.

Please also ensure you report the date at which the ethics committee approved the study as well as the complete date range for patient recruitment and follow-up in the Methods section of your manuscript.

" NMI study received funding from Universiti Kebangsaan Malaysia Research Grant

The active treatment and placebo were provided free of charge by study sponsor (B-CROBES SDN BHD).

The funders had no role in study design, data collection and analysis, decision to publish or in the preparation of the manuscript"

We note that you received funding from a commercial source: B-CROBES.

Reviewers' comments:

Reviewer's Responses to Questions

**Comments to the Author**

1. Is the manuscript technically sound, and do the data support the conclusions?

Reviewer #1: Partly

Reviewer #2: Yes

Reviewer #3: Partly

Reviewer #4: Yes

2. Has the statistical analysis been performed appropriately and rigorously? 

Reviewer #1: No

Reviewer #2: Yes

Reviewer #3: No

Reviewer #4: Yes

3. Have the authors made all data underlying the findings in their manuscript fully available?

Reviewer #1: Yes

Reviewer #2: Yes

Reviewer #3: No

Reviewer #4: Yes

4. Is the manuscript presented in an intelligible fashion and written in standard English?

Reviewer #1: No

Reviewer #2: Yes

Reviewer #3: Yes

Reviewer #4: Yes

5. Review Comments to the Author

Reviewer #1: There are a number of issues with the statistical analysis and sample size descriptions.

1. The authors first state the sample size is computed for type I error 0.01 and 80% power and then 1 sentence later say it was computed with type I error rate 0.05 and 90% power. Which?

2. The authors use reference 15 to justify sample size assumptions? What assumptions? What is a clinically relevant effect size here? What type of statistical test is this computation based on?

3. While the authors talk about a binary constipation outcome, I am seeing 9 different ordinal outcomes, with no adjustment for multiplicity.

4. The authors indicate that they will need 30 in each group with a minimum of 25, yet they have only 22 in one group. Again, I question these numbers anyway, given the number of hypothesis tests done for the primary outcome.

5. The conclusions should state whether the assumptions of your sample size computation were realized in the trial.

There are numerous minor problems with verb tenses, plurals and singulars, and definite and indefinite articles that could be corrected easily by an editor.

Reviewer #2: Dear authors,

Congratulations on your work with an unprecedented theme in the literature.

My considerations aim to encourage a reflection about the article, which will be scored below:

1. The term used "multi-strain probiotic and a prebiotic fiber" would mean symbiotic supplementation. I suggest that the authors use the term symbiotics, as it will define with greater accuracy the supplement offered in the essay.

2. The applied methodology is adequate and relevant, as well as the statistical analysis. However, when studying constipation, an analysis of food consumption and water consumption is essential, as this direct influence on the presence and severity of constipation is known. Another methodology that could be applied to estimate severity is the application of the bristol scale.

3. Regarding the discussion, I believe it can be deepened based on the scientific bases described in the literature on the pathophysiological mechanisms of constipation.

4. Regarding the limitations of the study, I believe it is important to explain the reasons for not being applied to assess water consumption and the Bristol scale.

Reviewer #3: General comments- this is an important study investigating the effects of a multis-train probiotic substance, added of FOS on the symptoms of constipation in persons living with PD. The study was well-designed, with a sample size calculation and blinding of all the personnel involved directly or indirectly with the study. However, I have some concerns which I will describe below.

1- Type of substance- to allow replicability of the study, and to open a deeper discussion of the data, it is essential to authors to describe the species’ strain number of the bacteria included in the substance

2- Is the substance used any type of commercial product? It is important to provide this information throughout the text.

3- With respect to the statistical analysis choice- why the authors used multiple t-test analysis (or correspondent non-parametric tests), instead of choosing a simultaneous analysis (ANOVA and their corresponding non-parametric test)? Performing multiple t-test analysis, besides the lack of information about time-group interaction, it increases the risk of Type-I error.

4- Considering the short duration of the study, and the number of dropouts, the authors would perform a sensitivity analysis, carrying forward the initial data of the patients who dropout. The existence of side-effects should not be neglected in the discussions. The use of FOS it is generally not well accepted by the elderly, due to the excessive gas formation, which can be an even greater complication in PD patients.

5- When discussing the limitation of the study due a lack of objective information on fiber intake, the authors could include the presence of other diet components also important to gut health, such as fatty acids, phytochemicals, and vitamins. All these components can serve as bacteria substrate and metabolites formation, which important to the gut functioning and metabolism.

6- How the authors found, from their study (in the abstract’s conclusion), the Type I evidence to the results? Type I evidence requires results from systematic reviews!

Reviewer #4: This is an interesting, well design clinical study evaluating the effect of 8 week supplementation of probiotics

in patients with Parkinson Disease suffering from constipation.

The data were designed, collected and recorded in accordance with CONSORT recommendation.

My comments are minor:

1) The name of probiotic strains used in this study are missing throughout the manuscript - please amend accordingly

2) Please discuss the potential bias in patients recruitment protocols based on Rome III criteria in light of current Rome IV criteria

3) Why Authors decided to chose Garrigues Questionnaire, as the low, positive likelihood ratios limit the clinical usefulness of this scale to identify individuals with constipation and there is no or scarce data on the reliability of the GQ or on its sensitivity to observe the change in bowel habits (McCrea et al. Review article: self-report measures to evaluate constipation. Aliment Pharmacol Ther 2008; 27(8): 638-48)

4) The results are promising but still the number of individuals included in this RCT is low, which limits its generalisability

6. PLOS authors have the option to publish the peer review history of their article (what does this mean?). If published, this will include your full peer review and any attached files.

Reviewer #1: No

Reviewer #2: No

Reviewer #3: No

Reviewer #4: No

---

## [Author Response · Author response to Decision Letter 0]

14 Nov 2020

Dear Editor and Reviewers,

Thank you for your helpful suggestions to improve the quality of the paper. We have uploaded point to point response to editorial and reviewers comments. We hope you find the manuscript acceptable for publication is PLOS One.

---

## [Decision Letter · Decision Letter 1]

15 Dec 2020

Multi-strain probiotics (Hexbio®) containing MCP®BCMC® strains improved constipation and gut motility in Parkinson’s disease:  A randomised controlled trial

PONE-D-20-21349R1

Dear Dr. Ibrahim,

We’re pleased to inform you that your manuscript has been judged scientifically suitable for publication and will be formally accepted for publication once it meets all outstanding technical requirements.

Kind regards,

Sandra Ribeiro

Guest Editor

PLOS ONE

Additional Editor Comments (optional):

The requests made in the first evaluation were met.

Reviewers' comments:

Reviewer's Responses to Questions

**Comments to the Author**

1. If the authors have adequately addressed your comments raised in a previous round of review and you feel that this manuscript is now acceptable for publication, you may indicate that here to bypass the “Comments to the Author” section, enter your conflict of interest statement in the “Confidential to Editor” section, and submit your "Accept" recommendation.

Reviewer #1: All comments have been addressed

Reviewer #2: All comments have been addressed

Reviewer #4: All comments have been addressed

2. Is the manuscript technically sound, and do the data support the conclusions?

Reviewer #1: (No Response)

Reviewer #2: Yes

Reviewer #4: Yes

3. Has the statistical analysis been performed appropriately and rigorously? 

Reviewer #1: (No Response)

Reviewer #2: Yes

Reviewer #4: I Don't Know

4. Have the authors made all data underlying the findings in their manuscript fully available?

Reviewer #1: (No Response)

Reviewer #2: Yes

Reviewer #4: Yes

5. Is the manuscript presented in an intelligible fashion and written in standard English?

Reviewer #1: (No Response)

Reviewer #2: Yes

Reviewer #4: Yes

6. Review Comments to the Author

Reviewer #1: (No Response)

Reviewer #2: (No Response)

Reviewer #4: The authors have addressed all comments enclosed in my initial review and this paper could be recommended for publication in PLOS ONE journal.

7. PLOS authors have the option to publish the peer review history of their article (what does this mean?). If published, this will include your full peer review and any attached files.

Reviewer #1: No

Reviewer #2: No

Reviewer #4: No

---

## [Editor Report · Acceptance letter]

17 Dec 2020

PONE-D-20-21349R1 

Multi-strain probiotics (Hexbio) containing MCP BCMC strains improved constipation and gut motility in Parkinson’s disease:  A randomised controlled trial 

Dear Dr. Ibrahim:

I'm pleased to inform you that your manuscript has been deemed suitable for publication in PLOS ONE. Congratulations! Your manuscript is now with our production department. 

Kind regards, 

on behalf of

Maria Lima Sandra Ribeiro 

Guest Editor

PLOS ONE